# Comparing Polyphosphate and Orthophosphate Treatments of Solution-Precipitated Aragonite Powders

**Boyang Gao** [1] **and Kristin M. Poduska** [1,2,*]

[1] Chemistry Department, Memorial University of Newfoundland and Labrador,
St. John's, NL A1C 5S7, Canada
[2] Physics & Physical Oceanography Department, Memorial University of Newfoundland and Labrador,
St. John's, NL A1B 3X7, Canada
[*] Correspondence: kris@mun.ca

**Abstract:** The aqueous and thermal stabilities of aragonite ($CaCO_3$) powders against phase conversion are important for industrial applications that rely on calcium carbonate. We describe the synthesis and characterization of solution-precipitated aragonite powders before and after exposure to different aqueous polyphosphate (SHMP) or orthophosphate ($PO_4$) treatments with concentrations ranging between 1–10 mM (∼1 g/L). Based on infrared spectra, differential scanning calorimetry, and thermogravimetric analyses, results show that orthophosphate treatments lead to secondary phase formation and complex thermal annealing behaviors. In contrast, polyphosphate treatments help to prevent against aragonite dissolution during water exposure, and also provide a slight increase in the thermal stability of aragonite with regard to conversion to calcite.

**Keywords:** calcium carbonate; orthophosphate; polyphosphate; infrared spectroscopy; differential scanning calorimetry; thermogravimetric analysis

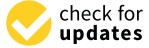



## 1. Introduction

The aqueous and thermal stabilities of aragonite ($CaCO_3$) powders against secondary phase formation are important for industrial applications that rely on calcium carbonate [1]. Surface treatments on calcium carbonate powders—whether aragonite or calcite—are commonly performed to prevent particle aggregation [2]. $CaCO_3$ precipitation and scaling represent formidable concerns in industrial processes, especially those that involve pipelines; unwanted deposits can reduce heat-transfer efficiency or accelerate corrosion, both of which have detrimental economic impacts [3,4].

Polyphosphate, in the form of the hexamer sodium hexametaphosphate (SHMP), is widely used to disperse calcium carbonate particles in industrial applications, and has also been used in laboratory sample preparations [5] and surface-charge measurements [6]. Although SHMP is widely used, it remains the subject of active study, with recent work highlighting its efficiency in inhibiting $CaCO_3$ scaling [7,8]. Other works report that SHMP mixed with liquid cement can accelerate the repair of concrete cracks by promoting anisotropic high-aspect-ratio aragonite crystal formation that is more effective in sealing cracks than the more blocky, isotropic calcite crystals [9]. Despite the widespread use of SHMP, its surface-specific interactions with calcium carbonate are poorly studied.

Orthophosphate salts, containing single phosphate units, are also widely used and studied for their interactions with calcium carbonate. For example, aragonite cement has been investigated as an interfacial material to accelerate bone tissue replacement and repair due to its higher solubility relative to calcite and calcium phosphates such as hydroxyapatite (HAp) [10,11]. In other instances, orthophosphate in waste water has been studied for its interactions with aragonite as a way to remove waterborne phosphate contaminants [12].

In this work, we compare the effects of aqueous polyphosphate and orthophophate treatments—with similar phosphate concentrations—on solution-precipitated aragonite

powders, including subsequent annealing treatments, while focusing on which conditions trigger secondary phase formation.

## 2. Materials and Methods

### 2.1. Precipitation

A solution-based precipitation reaction [13] was employed via stirring and temperature control to minimize phase heterogeneity in the final product. Starting solutions were both held at 90 °C: 60 mM $Na_2CO_3$ (ACS reagent grade (Sigma Aldrich, Oakville, ON, Canada) salt in ultrapure water (18.2 M$\Omega$·cm, 25 mL with starting pH 10.9) and 60 mM $CaCl_2$ (ACS reagent grade (Sigma Aldrich, Oakville, ON, Canada), 25 mL with starting pH 7.2), drop-by-drop over a 3 min time span while stirring at 400 rpm, to yield a white precipitate. The suspension cooled to room temperature, and was then centrifuged (4000 rpm for 10 min) and filtered to yield a supernatant (pH 8.7) and white solids, which were dried at ambient temperature for 12 h.

### 2.2. Annealing

In some experiments, the as-precipitated (untreated) powders were annealed in a pre-heated 400 °C furnace for 20 min.

### 2.3. Aqueous Treatments

After precipitation, some powders were treated in solutions with either polyphosphate or orthophosphate solutions. Since pH is one of the key factors that determines phosphate speciation in solution [14], treatments used different starting pH values (ranging from 6.5 to 10.5) and different phosphate concentrations (1–10 mM).

For polyphosphate treatments, we used $Na_6[(PO_3)_6]$ (sodium hexametaphosphate, (SHMP), ACS reagent grade, Alfa Aesar, Tewksbury, MA, USA) solutions at 1, 5, or 10 mM concentrations, with an initial pH value of 6.8. For some treatments, the pH was adjusted to 7.0 or 10.5 with NaOH (ACS reagent grade).

For orthophosphate treatments, we used standard phosphate buffer salts, but at concentrations and initial pH values similar to the SHMP treatments described above. The near-neutral solutions used $K_2HPO_4$ (Fisher Biotec, Burlington, ON, Canada, ACS reagent grade) and $KH_2PO_4$ (J.T.Baker, Radnor, PA, USA, ACS reagent grade); pH 7 used 5.3 mM and 4.6 mM, respectively, while pH 8 used 9.3m M an 0.65 mM, respectively). The more alkaline solutions used a mixture of $K_2HPO_4$ and $K_3PO_4$ (BDH chemicals, Radnor, PA, USA, ACS reagent grade); pH 9.5 used 9.5 mM and 0.5 mM, respectively, while pH 10.5 used 6.9 mM and 3.1 mM, respectively. The amounts above refer to the 10 mM orthophosphate treatment solutions; some experiments used diluted solutions at 5 mM and 1 mM.

For all treatments, suspensions used 0.1 g of powder in 10 mL solution for up to 3 weeks in capped glass-sample vials, without stirring. At regular intervals, aliquots of suspension were extracted and dried, to track any phase conversion. In some experiments, a pH meter was inserted in the supernatant to record pH at different time intervals.

### 2.4. Characterization

For bulk powder characterization, all measurements used an attenuated total reflectance (ATR) Fourier transform infrared (FTIR) spectrometer (Bruker Alpha-P, Billerica, MA, USA, collected with a single 45 degree reflection from a diamond ATR crystal, 400–4000 cm$^{-1}$ with 2 cm$^{-1}$ resolution). The air-dried powders were hand ground with an agate mortar and pestle before measurement. To track polymorph conversion, we used a qualitative comparison of relative intensities of the aragonite and calcite $\nu_2$ peaks. This method is not rigorously quantitative, but provides a useful metric for relative comparisons. These phase identifications are based on comparisons with mineral spectral libraries [15] We note that, in this work, ATR-FTIR enables identification of poorly crystalline and amor-

phous phases, which would not be feasible with some other structural characterization methods, such as X-ray diffraction.

Powder grains were imaged via scanning electron microscopy (SEM, FEI MLA 650F, FEI, Hillsboro, OR, USA) using secondary electron images to assess grain morphologies. To obtain high-resolution images, all powders were mounted on carbon tape and gold coated for better SEM conductivity. Energy dispersive X-ray (EDX, FEI MLA 650F, FEI, Hillsboro, OR, USA) analysis provided evidence of phosphorous incorporation.

To assess heat-related changes, we used differential scanning calorimetry (DSC, Mettler-Toledo DSC1, Mississauga, ON, Canada) measurements with ∼3.5 mg of sample, weighed into an aluminum crucible, placed in a pre-heated chamber (320 °C), and then heated from 320 to 550 °C at 10 °C/min under flowing $N_2$ (50 mL/min). To support these DSC data, additional experiments characterized the phase composition before and after heating at 440 °C (pre-heated furnace) for 20–30 min, followed by an air quench. For thermogravimetric analysis (TGA, TA Instruments Q500, TA Instruments, New Castle, DE, USA), we used 8 mg of powder for each scan, with nitrogen flowing (50 mL/min) throughout the analysis. The temperature range spanned between 20 °C and 100 °C, with a heating ramp of 10 °C/min.

## 3. Results

### 3.1. Before Aqueous Treatments

Based on ATR-FTIR spectra (Figure 1a), the as-precipitated (untreated) powders show typical vibration peaks of aragonite, with a $v_3$ peak (1400 cm$^{-1}$), $v_1$ peak (1082 cm$^{-1}$), $v_2$ (858 cm$^{-1}$), and $v_4$ doublet (712 and 700 cm$^{-1}$) [16,17]. We note that these conventional peak labels correspond to specific vibrational modes of the carbonate moieties: $v_3$ is an asymmetric stretch, $v_1$ is a symmetric stretch, $v_2$ is an out-of-plane bend, and $v_4$ is an in-plane wag. For clarity, the ATR-FTIR spectra in each Figure bear explicit labels for the aragonite $v_2$ and calcite $v_2$ peaks. (Full-range FTIR-ATR spectra (400–4000 cm$^{-1}$) can be found in Supplementary Material Figure S1). SEM images (Figure 1g) show that these powder grains bear an acicular morphology, which is also typical of aragonite.

We used ATR-FTIR, supported by SEM imaging, to track the phase stability of as-precipitated powders following water exposure after annealing (0.1 g powder in 10 mL ultrapure water). Without annealing, there was no evidence of a calcite $v_2$ peak after up to three weeks of water exposure (Figure 1b,h). After annealing the as-precipitated powders (400 °C for 20 min), a calcite-related shoulder appears at 875 cm$^{-1}$ (Figure 1e). Despite this secondary phase, the crystal habits of the annealed powder grains appear highly similar to the unannealed grains (Figure 1i). We note that at lower annealing temperatures (350 °C), no calcite formed even after 3 hours of heat treatment (SEM images of 350 °C-annealed powders are shown in Supplementary Material Figure S2).

After 1 week of exposure to water (0.1 g in 10 mL nanopure water), there are significant changes to the annealed powders in terms of the relative aragonite:calcite ATR-FTIR peak intensities (Figure 1f) and the crystal habits (Figure 1j). The ATR-FTIR data show a more pronounced $v_2$ calcite peak, relative to aragonite $v_2$. SEM images show many small rhombohedral crystallites that form on the needle faces, consistent with the typical calcite crystal habit (Figure 1g).

Based on these experiments, we observe that:

(1) Aragonite alone, whether as-precipitated or subjected to a lower temperature (350 °C) anneal, has good phase stability in water over a span of one week.

(2) Aragonite annealed at a slightly higher temperature (400 °C) introduces calcite as a secondary phase. Subsequent exposure to water triggers more calcite formation.

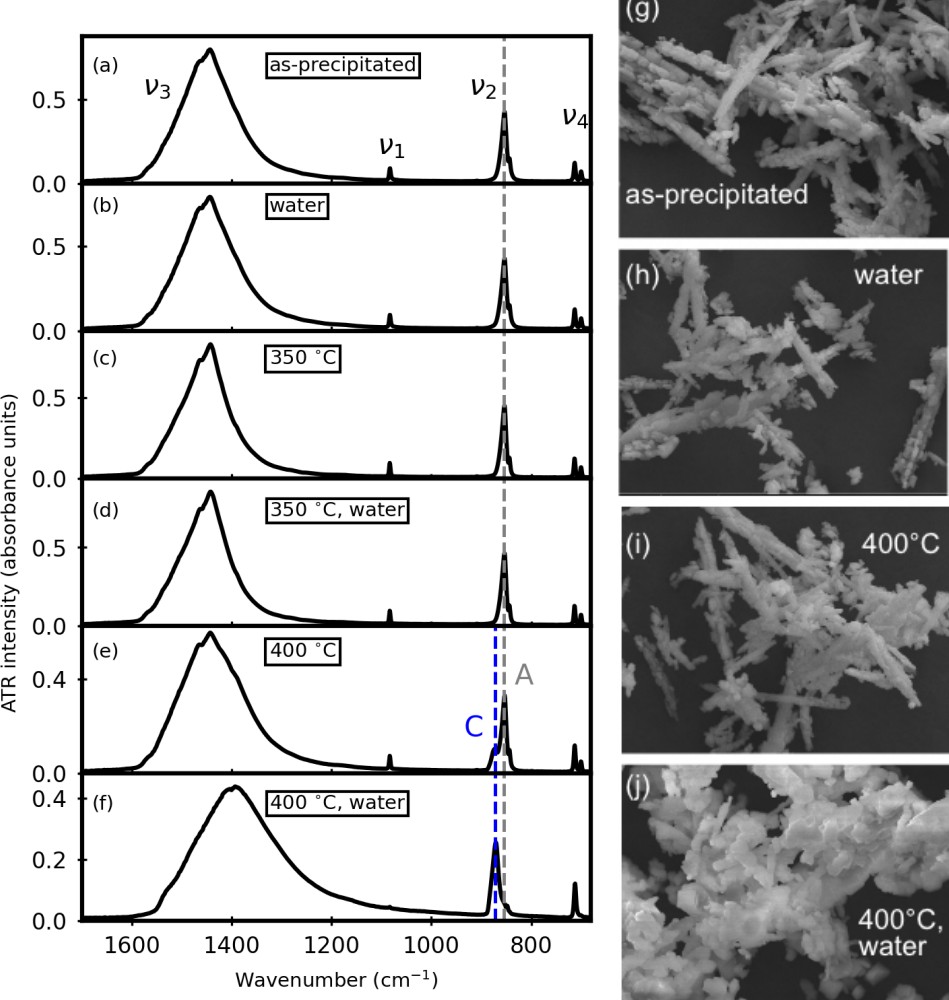

**Figure 1.** Representative ATR-FTIR spectra for (**a**) as-precipitated powders, (**b**) as-precipitated powders after 1 week in water, (**c**) as-precipitated powders + 350 °C anneal, (**d**) as-precipitated powders + 350 °C, then 1 week in water, (**e**) as-precipitated powders + 400 °C anneal, and (**f**) as-precipitated powders + 400 °C anneal, followed by 1 week in water. The grey vertical line (858 cm$^{-1}$, labelled A) corresponds to the aragonite $\nu_2$ peak, while the blue vertical line (875 cm$^{-1}$, labelled C) shows the calcite $\nu_2$ peak. Representative SEM images for (**g**) as-precipitated powders, (**h**) as-precipitated powders after exposure to water (1 week), (**i**) annealed, (**j**) annealed then exposed to water (1 week). Each image covers a width of 50 μm.

### 3.2. After Aqueous Treatments

We explored two different groups of aqueous phosphate treatments: polyphosphate (SHMP) solutions and orthophosphate (PO$_4$) solutions. Investigations involved different starting pH values (6.5–10.5) as well as different phosphate concentrations (1, 5, or 10 mM), and all utilized 0.1 g of powder in 10 mL of treatment solution.

For SHMP, whether unannealed or annealed starting powders, the ATR-FTIR spectra showed no evidence of phosphate-containing secondary phases, even after up to three weeks of immersion in SHMP (1, 5, or 10 mM). Figure 2a,b representative spectra, which only have peaks due to aragonite (in the case of as-precipitated powders, for comparison with Figure 1a,b) or a mixture of aragonite and calcite (in the case of 400 °C-heated powders, for comparison with Figure 1e,f). Furthermore, the relative aragonite:calcite peak intensities remain consistent before and after SHMP treatment, which suggests, qualitatively, that their relative fractions within the sample remain similar. (Full-range FTIR-ATR spectra are shown in Supplementary Material Figure S3). SEM images (Figure 2d,e) show that annealing to 400 °C does not alter the crystallite morphologies.

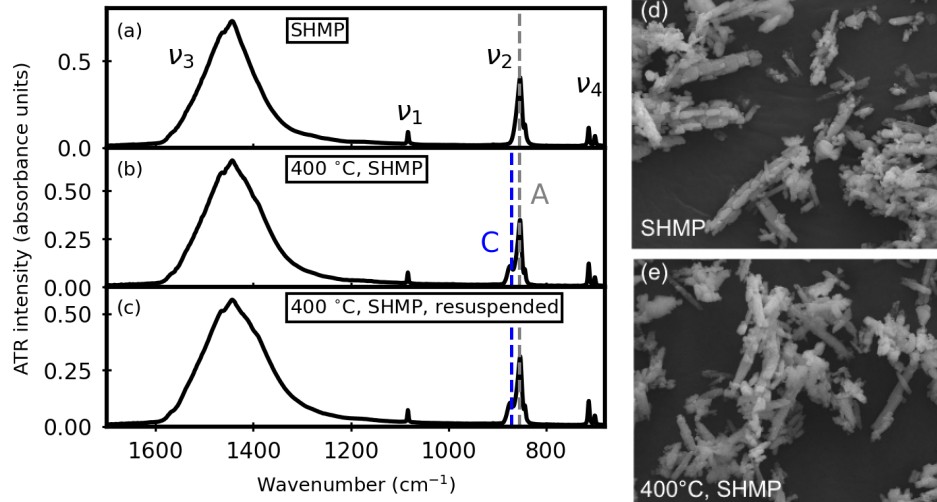

**Figure 2.** Representative ATR-FTIR spectra (**a**–**c**) and SEM images (**d**,**e**) for polyphosphate-treated powders (1-week suspension in 10 mM SHMP). The grey vertical line (858 cm$^{-1}$, labelled A) corresponds to the aragonite $\nu_2$ peak, while the blue vertical line (875 cm$^{-1}$, labelled C) shows the calcite $\nu_2$ peak. In all cases, the ATR-FTIR data show spectra comparable to the starting materials, with no additional peaks, and no discernable change in the relative aragonite:calcite peak intensities (compare with Figure 1a,e). The SEM images show that annealing to 400 °C does not cause any discernable differences to the crystallite morphologies. Each image covers a width of 50 μm.

Since the SHMP-treated mixed-phase (400 °C) powders demonstrated very little dissolution during treatment, we checked to see whether the polyphosphate treatment created a robust change to the powder by removing it from suspension, drying, and then suspending in water. ATR-FTIR spectra of these re-suspended powders (Figure 2c) show consistent aragonite:calcite $\nu_2$ IR peak intensity ratios, suggesting that no further calcite formation occurred in these resuspended powders. We note that EDX data confirmed the presence of phosphorus uniformly on the SHMP-treated powders, although the corresponding ATR-FTIR spectra did not show any phosphate-related peaks. Furthermore, although the data shown in Figure 2 correspond to SHMP treatments with initial pH 6.5, outcomes were the same with pH-adjusted treatments at 7.0 and 10.5. (These data are provided in Supplementary Material Figure S4). Thus, all SHMP treatments appear to adhere to the powder, and provide protection against aragonite-to-calcite conversion in water.

To explore whether other kinds of phosphate additives have a similar effect, we investigated orthophosphate (PO$_4$) additives. However, orthophosphate treatments trigger significant amounts of secondary phosphate minerals, as shown by the ATR-FTIR data in Figure 3 (Full-range FTIR-ATR spectra are shown in Supplementary Material Figure S5). For the near-neutral (pH < 8.0) and the more alkaline (pH > 9.5) treatment conditions, hydroxyapatite (HAp, Ca$_5$(PO$_4$)$_3$(OH)) was a dominant secondary phase. For intermediate pH values (pH 8 and pH 9.5), some brushite (dicalcium phosphate dihydrate (DCPD), CaHPO$_4\cdot$2H$_2$O) also formed.

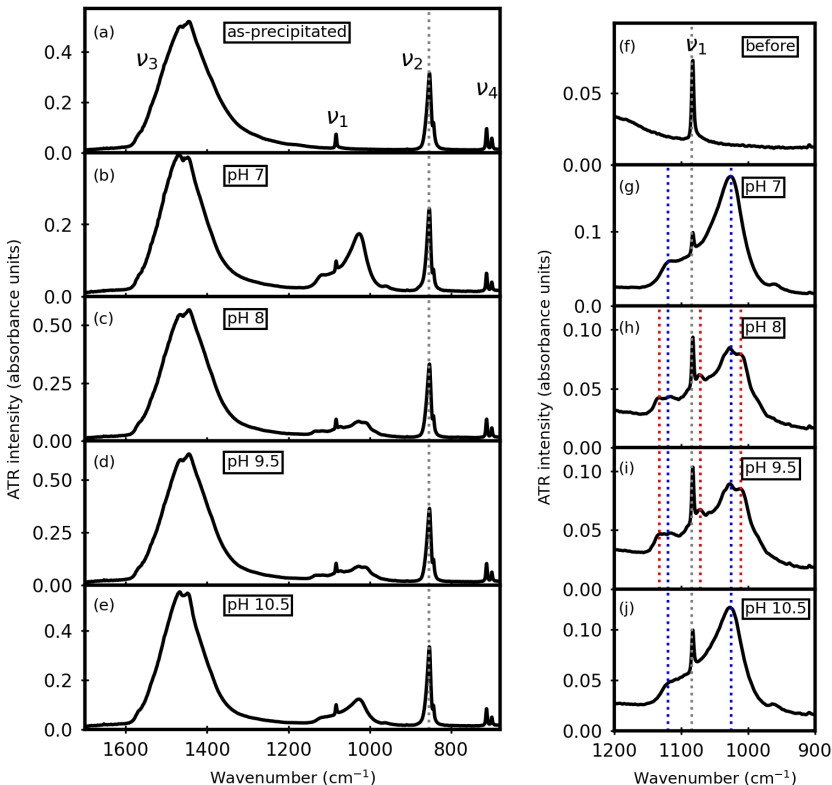

**Figure 3.** Representative ATR-FTIR data of as-precipitated powders before (**a**) and after orthophosphate treatments beginning near-neutral (pH 7 in (**b**)) and progressing to more alkaline (pH 8 in (**c**), pH 9.5 in (**d**), and pH 10.5 in (**e**)). Evidence of secondary phosphate mineral formation (900–1200 cm$^{-1}$) is emphasized in the zoomed panels (**f**–**j**). The grey vertical lines with band labels correspond to aragonite peaks, while the blue vertical lines and red vertical lines denote the peak positions of HAp and brushite, respectively.

### 3.3. Time Trends vs. pH during Aqueous Treatments

Since phosphate additives have buffering capacity, we monitored pH during the aqueous treatments. Figure 4a shows pH *vs.* time trends for a representative SHMP set of samples (analogous to data in Figure 2: annealed samples in water only, during SHMP treatment, and after SHMP treatment during re-suspension in water). These data show a qualitatively similar pH trend over time, with two distinct pH regimes: a fast rise in the first hour, and then a steady decline thereafter.

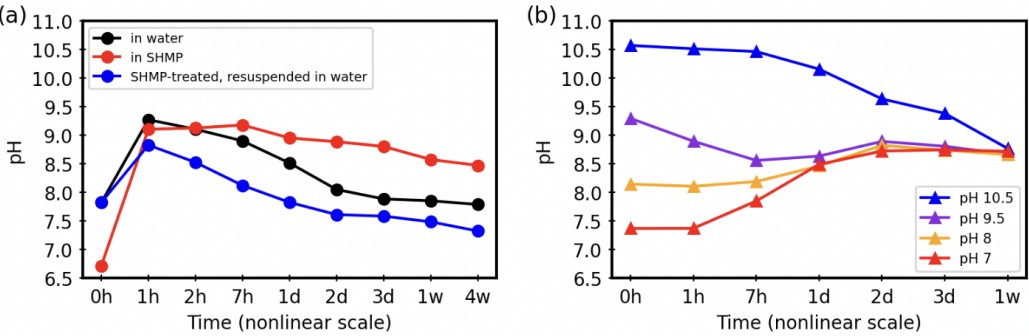

**Figure 4.** Representative pH *vs.* time trends for (**a**) SHMP-treated samples (annealed, SHMP-treated, and re-suspended SHMP-treated powders; see also Figure 2) and (**b**) orthophosphate-treated samples (different starting pH values, all at 10 mM; see also Figure 3).

The pH trends during orthophosphate treatments are quite different. Figure 4b shows trends for a set of samples analogous to those described in Figure 3 (starting pH values ranging from 7.5 to 10.5, all at 10 mM). It is clear that none of these orthophosphate solutions act as true buffers while the powders are suspended. After 1 week, all solutions converge to a pH value of 8.6.

We are cautious in interpreting these pH trends since this system is complex. When the solid $CaCO_3$ is added to water, we expect a small amount of dissolution based on the known solubility constants for calcite and aragonite. Dissolution of ambient $CO_2$ in the water also affects the ionic composition of the solution over time. Both of these factors contribute to changes in the point of zero charge (PZC) for calcium carbonate particles, due to the influence of ions in and near the particle surfaces [18].

### 3.4. Thermal Stability after Aqueous Treatments

We also followed the phase behavior when phosphate-treated powders were subjected to thermal treatments, using differential scanning calorimentry (DSC) to look for endothermic peaks related to phase changes. We expect a thermally induced aragonite-to-calcite transition, based on studies by other researchers [19,20]. Figure 5 shows DSC scans at 10 °C/min, where endothermic peaks are visible in both treated and untreated samples, with comparable peak areas. However, the DSC peak occurs at higher temperatures in SHMP-treated samples (465 °C, regardless of SHMP treatment concentration) compared to the untreated sample (435 °C). We note that the peak area from the as-precipitated aragonite is slightly larger than that of SHMP-treated samples. Our data do not provide a conclusive explanation for this observation; however, we speculate that it could be related to phase changes involving the phosphate.

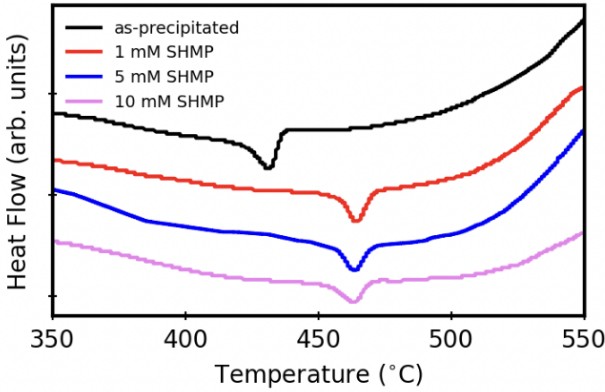

**Figure 5.** Representative DSC traces for as-prepared and SHMP-treated powders, all at 10 °C/min heating. Endothermic peaks point downward. Data curves are offset along the vertical axis for clarity.

To confirm that the endothermic DSC peak is due to an aragonite-to-calcite phase conversion, we used ATR-FTIR to assess the phase composition of the samples following DSC measurement. Figure 6 shows that the DSC heating profile caused complete conversion to calcite as-precipitated and SHMP-treated samples; after heating, the aragonite $\nu_1$ and lower wavenumber $\nu_4$ peaks disappear completely, and the $\nu_2$ peak shifts to a slightly higher wavenumber. For the SHMP-treated sample (10 mM), there is also a broad rise in the baseline between 1000 and 1200 cm$^{-1}$, which is consistent with what others have reported for poorly crystallized phosphate phases [15]. Therefore, heating SHMP-samples to sufficiently high temperatures can trigger both aragonite–calcite conversion and phosphate secondary phase formation. Even so, SEM images of powders after DSC measurements (Figure 6b) do not show any appreciable crystal morphology changes, maintaining their acicular crystal habits. Research by others [21] shows that calcite-after-aragonite pseudomorphs can form, wherein calcite forms after annealing aragonite, but the crystal morphologies re-

main unchanged. This means that SEM was not effective in tracking heat-induced phase transformations since calcite appears in an atypical pseudomorphic crystal shape.

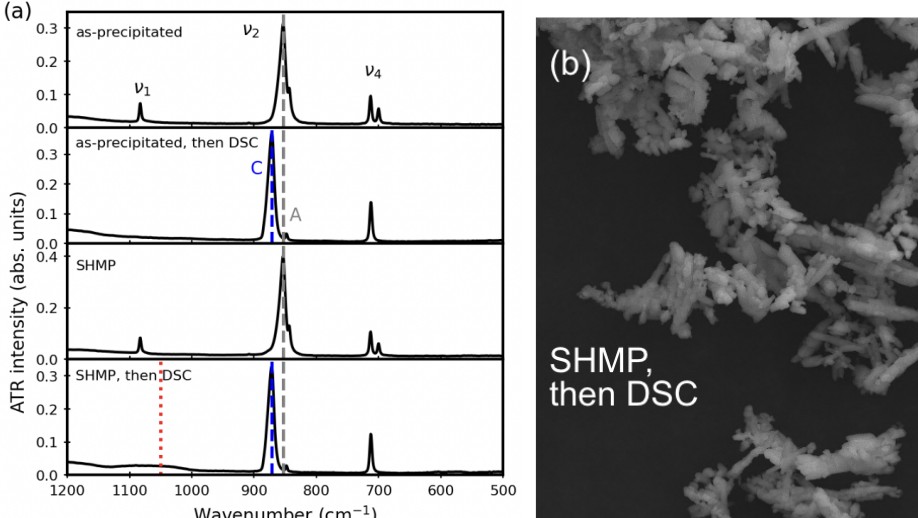

**Figure 6.** (**a**) Representative ATR-FTIR spectra show that after DSC measurements, both the as-precipitated and SHMP-treated powders convert entirely to calcite (no $\nu1$ peak, and only one $\nu4$ peak). The grey vertical line (858 cm$^{-1}$, labeled A) corresponds to the aragonite $\nu_2$ peak, while the blue vertical line (875 cm$^{-1}$, labeled C) shows the calcite $\nu_2$ peak. The SHMP-treated sample (10 mM) also shows a very broad peak in the 1000–1200 cm$^{-1}$ region (indicated by the red-vertical-dotted line), suggestive of a small phosphate secondary phase [15]. (**b**) Representative SEM image of SHMP-treated aragonite after DSC measurement, covering a width of 50 μm.

In separate experiments, we annealed SHMP-treated aragonite at 440 °C for 30 min. The higher temperature and longer annealing time provided even stronger evidence for phosphate remaining on the powders after SHMP treatments. ATR-FTIR data in Figure 7a show that higher SHMP solution concentrations during treatment cause the highest phosphate-related hump (1000–1200 cm$^{-1}$). We also note that the untreated sample has no aragonite remaining after heating, while the SHMP-treated samples all bear some remaining aragonite peak intensity after heating.

For comparison, we also annealed the orthophosphate-treated aragonite under the same conditions (440 °C for 30 min). Representative ATR-FTIR data show that for samples that had predominant HAp secondary phase before heating, the annealing treatment did not cause a significant change in the HAp peaks. However, these samples did show decreased aragonite peak intensities and increased calcite peak intensities; as an example, the top two panels of Figure 7b show samples treated at pH 7 (10 mM) before and after heating. For samples that had both HAp and brushite before heating, the annealing treatment caused thermal conversion of aragonite to calcite, as well as thermal coversion of brushite to HAp. As an example, the bottom two panels of Figure 7b show samples treated at pH 8 (10 mM) before and after heating. We note that there are also changes in the relative intensities of the calcite and aragonite peaks, but our ATR-FTIR data cannot provide a definitive answer to whether aragonite is consumed to form the phosphate-bearing minerals at the expense of calcite, or if the phase transition to calcite is an intermediate step.

To analyze the evolution of the phase composition changes in the orthophosphate-treated samples in more detail, we used thermogravimetric analyses (TGA) from 20 to 800 °C. Figure 8 compares the changes in % mass, at 10 °C/min sweep rate, of powders treated with orthophosphate at different starting pH values (all at 10 mM). The full temperature range (Figure 8a) shows qualitatively similar mass changes in all samples, with small mass losses below 600 °C and a steep mass loss near 700 °C.

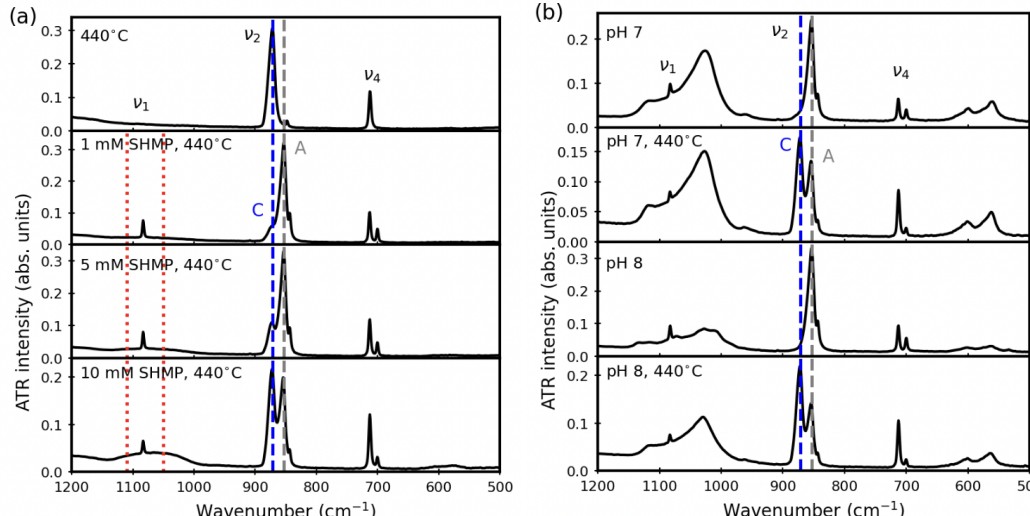

**Figure 7.** Representative ATR-FTIR data for samples annealed at 440 °C after aqueous treatment in either (**a**) SHMP (1, 5, 10 mM) or (**b**) orthophosphate (pH 7 or pH 8, both at 10 mM). The grey vertical line (858 cm$^{-1}$, labeled A) corresponds to the aragonite $\nu_2$ peak, while the blue vertical line (875 cm$^{-1}$, labeled C) shows the calcite $\nu_2$ peak. The red-dotted lines indicate poorly crystalline phosphate secondary phases.

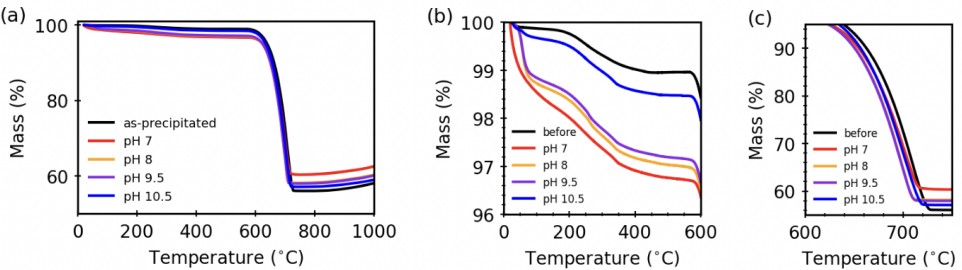

**Figure 8.** Representative TGA traces compared between as-precipitated and orthophosphate-treated powders during 10 °C/min heating. The full temperature range is shown in (**a**), while (**b**,**c**) zoom in on the low-temperature and high-temperature transition regions, respectively.

Observing the low-temperature range more closely (Figure 8b), the pH 10.5 sample is most similar to the as-precipitated sample, with plateaus at 100–200 °C and from 400–600 °C, for a total mass loss in the range of 1%. In contrast, Figure 8b also shows that there are larger mass losses for samples treated in the pH 7–9.5, range, with no clear plateaus and a total mass loss near 3%. For the higher TGA temperature range (Figure 8c), all samples show similar rapid mass losses beginning near 575 °C and ending by 750 °C, leaving only 58–63% of the total weight in the solid. Values for these changes are compared in Table 1.

The large mass loss near 700 °C is consistent with calcination, whereby CaCO$_3$ transforms to CaO as CO$_2$ is released [22]. Based on this, we calculate the ideal % mass loss due to calcination. We then assume that the difference between this ideal mass loss (due to full calcination) and the observed mass loss (Figure 8) are due to calcium phosphate secondary phases, which gives us an estimate of % mass of secondary phases and/or uncalcined CaCO$_3$ that remains in each sample. A summary of these estimates appears in Table 1.

**Table 1.** Summary of % mass losses during TGA measurements for powders with orthophosphate treatments at different starting pH values.

| Sample/Mass (%) | Low T Loss (20–575 °C) | High T Loss (575–750 °C) | Assumed CaO Mass Remaining | Excess Mass Remaining |
| --- | --- | --- | --- | --- |
| ideal | 0.0% | 43.0% | 57.0% | 0.0% |
| as-precipitated | 1.1% | 40.9% | 54.2% | 3.8% |
| pH 10.5 | 1.6% | 39.4% | 52.2% | 6.8% |
| pH 9.5 | 2.9% | 37.0% | 49.0% | 11.1 % |
| pH 8 | 3.0% | 36.7% | 48.6% | 11.7 % |
| pH 7 | 3.3% | 34.2% | 45.3% | 17.2 % |

## 4. Discussion

Here, our work contrasts the differences between polyphosphate and orthophosphate treatments, as applied to solution-precipitated aragonite powders. Polyphosphate treatments not only help to prevent against aragonite dissolution during exposure to water, but also provide a slight increase in the thermal stability of aragonite with regard to conversion to calcite. In contrast, orthophosphate treatments trigger secondary phases to form during the treatments, which leads to more complex thermal annealing behaviours.

In the aqueous treatments described herein, we used comparable phosphate concentrations (1–10 mM) between the polyphosphate and orthophosphate solutions. As a result, our orthophosphate treatments were not designed to be effective buffers, even though they used standard buffer salts. To explain this in more detail, the normal range for potassium phosphate buffers based on $K_2HPO_4$ and $KH_2PO_4$ is pH 6–8, based on the Henderson–Hasselbalch Equation [23] (Equation (1)):

$$pH = pKa + log([base]/[acid]). \tag{1}$$

In this case, $KH_2PO_4$ functions as the acid while $K_2HPO_4$ is the conjugated base. To achieve buffering at higher pH values, $K_2HPO_4$ becomes the acid and one must use a different salt, $K_3PO_4$, as the conjugated base. In general, a buffer is effective only when the base:acid ratio is in the range of 0.1–10, with the best buffering at a ratio of 1. Our orthophosphate treatments bear ratios that are not unity: some are within the useful range (1.1 for pH 7 and 0.4 for pH 10.5), but others are well outside (15 for pH 8 and 0.05 for pH 9). Therefore, it is not surprising that Figure 4 shows that none of the orthophosphate treatments function as true buffers while the powders are in suspension. Furthermore, we note that Figure 4 shows pH *vs.* time data for suspensions with the highest orthophosphate concentrations of this study (10 mM); the diluted treatments (1 mM, 5 mM) have even less buffering capacity.

Our TGA data (Figure 8 and Table 1) provide a useful complement to the secondary phase information we gathered from ATR-FTIR spectral data (Figure 3). Mass loss at lower temperatures may be due to loss of structural water [24]. However, we see that even in the as-precipitated sample, there is 1% mass loss below 575°. Furthermore, the pH 7 sample, which showed only HAp and not brushite, shows the highest mass loss below 575 °C (just above 3%). Based on these data, it is likely that the low T mass losses are not due solely to structural water.

For the high-temperature mass loss (575–750 °C) that we attribute to $CaCO_3$ calcination, we see in Table 1 that even the as-precipitated sample has incomplete calcination (3.8% excess mass). The excess mass increases as the pH of the orthophosphate treatment decreases, reaching a maximum of 17% for the pH 7 treatment. For the orthophosphate-treated samples, the ATR-FTIR spectra in Figure 7b confirm that brushite converts to HAp after high-temperature annealing; work conducted by others shows that the thermal decomposition temperature for HAp occurs at temperatures above our TGA data range

(1000–1360 °C [25]). Thus, we infer that the excess mass likely includes a combination of uncalcined $CaCO_3$, HAp, and monetite ($CaHPO_4$, anhydrous form of brushite [26]).

Research conducted by others provides support for our pH-dependent trends in secondary phase formation. While $\mu$M concentrations are often too low to trigger significant phosphate mineral formation in the presence of $CaCO_3$ [27,28], higher (mM) phosphate concentrations have been shown to cause phosphate mineralization [29]. In acidic and neutral environments, the strong dissolution of calcium carbonate contributes to the interaction between calcium and phosphate ions, resulting in calcium phosphate precipitates [30]. However, as pH increases, phosphate uptake gradually decreases due to inhibition of calcium carbonate dissolution; this is consistent with the excess mass trend we see from TGA data (Table 1), where the excess mass (due in part to phosphate uptake) is largest for the most acidic orthophosphate treatment (pH 7).

With respect to the formation of phases, $HPO_4^{2-}$ ion is ubiquitous at pH 7–10 and this range often produces brushite. Outside of this range, HAp is the dominant phase [31], which is consistent with our findings of brushite only after pH 8.0–9.5 orthophosphate treatments.

Comparing the phase complexity of the orthophosphate-treated powders to the almost-unchanged SHMP-treated powders, it is apparent why polyphosphate is widely used as a dispersant and anti-flocculant. It is worth noting that our 10 mM treatments are equivalent to 6 g/L, which is a concentration orders of magnitude higher compared to other recent studies [32]. Even with the high concentrations, we find that any secondary phosphate phases are barely detectable with ATR-FTIR spectra, even after annealing. However, its surface-specific interactions with calcium carbonate are poorly studied.

## 5. Conclusions

Our work compares aqueous polyphosphate and orthophosphate treatments on aragonite, using similar phosphate concentrations throughout. Polyphosphate treatments not only help to prevent against aragonite dissolution during exposure to water, but also provide a slight increase in the thermal stability of aragonite concerning the conversion to calcite. In contrast, orthophosphate triggers secondary phases to form during the treatments, which leads to more complex thermal annealing behaviours.

**Supplementary Materials:** The following supporting information can be downloaded at: https://www.mdpi.com/article/10.3390/solids3040042/s1, Figure S1: Full-range ATR-FTIR data to complement Figure 1; Figure S2: Additional SEM images to complement Figure 1; Figure S3: Full-range ATR-FTIR data to complement Figure 2; Figure S4: ATR-FTIR data to complement Figure 3; Figure S5: Full-range ATR-FTIR data to complement Figure 5.

**Author Contributions:** Conceptualization, methodology, writing—original draft preparation, review and editing: B.G. and K.M.P.; investigation and data curation: B.G.; supervision, project administration, and funding acquisition: K.M.P. All authors have read and agreed to the published version of the manuscript.

**Funding:** This research was funded by the Natural Science and Engineering Research Council of Canada (NSERC) grant number 2018-04888.

**Data Availability Statement:** The data presented in this study are available on request from the corresponding author.

**Acknowledgments:** J.B. Lin (ICP-OES, DSC, and TGA at the Centre for Chemical Research and Training) and the Microanalysis Facility (SEM/EDX) of Memorial University's CREAIT network for access to characterization facilities. A. Schmidt and S. Kababya (Technion, Israel Institute of Technology) for insightful conversations.

**Conflicts of Interest:** The authors declare no conflict of interest. The funders had no role in the design of the study; in the collection, analyses, or interpretation of data; in the writing of the manuscript; or in the decision to publish the results.

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
