# Peer review of "Comparing Polyphosphate and Orthophosphate Treatments of Solution-Precipitated Aragonite Powders"

_solids, doi:10.3390/solids3040042_

Round 1
Reviewer 1 Report
The manuscript is well-written and well-organized. Major conclusions are well supported by the experimental data
The manuscript covers important topic of chemical stability of aragonite phase and provides detailed analysis of chemical stability under different treatments. The research is new. It is important for many practical applications of aragonite, especially in biomedical field. The methodology is clear and well suited for the purpose of this research. The conclusions are well supported by the experimental data. References are appropriate. The quality of figures is goodAuthor Response
Reviewer 1 did not suggest any changes to the manuscript.
Reviewer 2 Report
The manuscript present an interesting topic of research work. However, there are some unclear or questionable statement. Addition data or information is needed.
1. Line 83, Why the pre-heat temperature (440C) is difference with the control experiment (400C).
2. Figure 1, suggest to add in SEM image for sample 350C and 350C,water.
3. Line 124, “The ATR-FTIR data show a 123 more pronounced ν2 calcite peak, relative to aragonite ν2.”, Label the v2(calcite) and v2(aragonite) peak in figure 1. Suggest use two different label v2-1 and v2-2 or others symbol. Since V2 data is use to differential calcite and aragonite, compared graph (zoom scale graph) is needed to added.
4. Line 129, “…has good phase stability in water over a span of several weeks.”, from Fig.1, experiment only conducted one week.
5. Line 143, “Furthermore, the relative aragonite:calcite peak intensities remain consistent before…”, intensities of peak is used to make comparison, it is suggest to prepare a table to show its value.
6. Line 154, “Furthermore, even though the data 154 shown in Figure 2 correspond to SHMP treatments with initial pH 6.5, outcomes were the 155 same with pH-adjusted treatments at 7.0 and 10.5.” , this statement obtained from which results?
7. Line 163, “For intermediate 163 pH values, some brushite (dicalcium phosphate dihydrate (DCPD), CaHPO4·2H2O) also 164 formed.”, from Figure, which peaks or results indicate formation of DCPD phase?
8. Line 171, “These data show a 170 qualitatively similar pH trend over time, with two distinct pH regimes: a fast rise in the 171 first hour, then a steady decline thereafter.” Why this is happening, any explanation?
9. Line 176, “After 1 week, all solutions converge to a pH 176 value of 8.6.” Why this is happening, any explanation?
10. Line 186, “We note that the peak area from the as-precipitated aragonite is slightly larger than that of SHMP-treated samples.”, How much, what is the value and why?
11. Line 194, “…which is suggestive of a small amount of poorly-crystallized phosphate secondary phase.” This statement is based on which prove, data or argument. Please provide reference if any.
12. Line 199, “This is not uncommon to have pseudomorphs form in the shape of aragonite, after annealing form calcite.” Refer paper [20], formation of pseudomorphs observed in nature high pressure marbles in Syros, Greece. In this work, no high pressure is used Please explain what happen in this case?
13. Line 230, “There are larger mass losses for samples treated in the pH 7-9.5, range, with no clear plateaus and a total mass loss near 3%.”, no clear explanation on statement that refer to Fig.8b.
14. Line 265, “Mass loss at lower temperatures may be due to the decomposition of the structural water, especially in phases such as brushite.”, any results indicate the existing of brushite in the sample? The following state there is no brushite “Furthermore, the pH 7 sample, which showed only HAp and not brushite, shows the highest mass loss below 575◦C (just above 3%).”
15. Line 275, “…the ATR-FTIR spectra confirm that secondary phosphate phases HAp and brushite are present…”, which ATR-FTIR spectra is refer to?
Reviewer 3 Report
In this paper, authors compared the effect of aqueous polyphosphate and orthophophate treatments on solution-precipitated aragonite powders, and focused on the conditions trigger secondary phase formation. The manuscript was written reasonably, and it could be accepted after some problems are addressed.
1. too many keywords and can not highlight the characteristics of the article, it is recommended to retain 5 - 6。
2. line 37, the sentence "Despite the widespread use of SHMP" is repeated and can be replaced by "However".
3. In section 2.3, the condition of annealing of some experiments were different from the control ones, why?
4. Please add the instrument model and manufacturer of EDX, DSC and TG
5. please give an ATR-FTIR spectra of 4000-400 cm-1.
6. Why are phase transition studies not performed using powder X-ray diffraction
Round 2
Reviewer 2 Report
Suggested label the v2a(calcite) and v2b(aragonite) peak in figure 1, 2 & 6 and others related figure. Please amend label in the text as well.
Author Response
In response to the Reviewer's comment, we have added wording to the main text to clarify that the origin of the peak labels is convention, and that the designation "nu2" indicates that the vibrational modes correspond to out-of-plane vibrations within the carbonate moiety. For this reason, rather than using an unconventional peak label (v2a and v2b), we have updated Figures 1, 2, 6, and 7 with clearer lines and explicit labels on the figure to denote which peak corresponds to calcite nu2 and which peak corresponds to aragonite nu2.